# The First Shall Be First: Letter-Position Coding and Spatial Invariance in Two Cases of Attentional Dyslexia

**DOI:** 10.3390/brainsci15090967

**Published:** 2025-09-06

**Authors:** Jeremy J. Tree, David R. Playfoot

**Affiliations:** Department of Psychology, Faculty of Medicine, Health and Life Sciences, Swansea University, Swansea SA2 8PP, UK; d.r.playfoot@swansea.ac.uk

**Keywords:** posterior cortical atrophy, attentional dyslexia, reading

## Abstract

**Background/Objectives:** Previous research has demonstrated that the initial letters of a word likely play a *privileged role* in visual word recognition, such that reading and visual recognition errors reflecting changes in this position are much less likely. For example, prior case studies of attentional dyslexia reported that participants were most accurate at rejecting nonwords formed by transposing a word’s *first two* letters (e.g., *WONER* from *OWNER*) compared to transpositions in later positions. The current study aimed to replicate and extend this finding in patients with posterior cortical atrophy (PCA), a neurodegenerative condition associated with visuospatial and attentional impairments. **Methods:** Two PCA patients completed lexical decision tasks involving five-letter real words and nonwords created either by transposing adjacent letters (in positions 1 + 2, 2 + 3, 3 + 4, or 4 + 5) or using matched nonword controls. To assess robustness, tasks were repeated across test–retest sessions. Stimuli were presented in both canonical horizontal and non-canonical vertical (marquee) formats. Accuracy, response bias, and sensitivity (d′) were estimated, with 95% confidence intervals derived from a nonparametric bootstrap procedure. Within-case logistic regressions were also conducted to illustrate the findings. **Results:** Both patients showed significantly higher accuracy and lower response bias for 1 + 2 transposition nonwords relative to other positions. This early-letter advantage persisted across test–retest observations and was maintained when words were presented in the vertical format, suggesting *orientation-invariant* effects. The bootstrap and regression analyses provided convergent support for these results. **Conclusions:** The findings provide novel evidence in PCA that the encoding of early letter positions operates independently of visual orientation and persists despite attentional deficits. This supports models in which the initial letters serve as a key *anchor point* in orthographic processing, highlighting the privileged and resilient status of early letter encoding in visual word recognition.

## 1. Introduction

Understanding how the spatial position of letters affects word recognition has been a focus of reading research across typical adult readers and neuropsychological populations. Converging evidence suggests not all letter-in-string positions are equal: letters in the first *two positions* may enjoy a privileged status in word recognition [1]. The present study explores this issue using two neuropsychological case studies of posterior cortical atrophy (PCA), a syndrome characterised by severe attentional impairments. We ask whether such a letter-position advantage in visual word recognition remains even in the face of these attentional deficits. Below, we summarise key findings on letter-position effects, highlighting studies of typical readers and key patient groups (attentional dyslexia and PCA), and discuss implications for models of orthographic encoding and spatial attention.

Studies of typical readers highlight that certain letter positions in words are especially important. For example, under brief presentations, readers benefit from primes containing only the initial or final letters of a word. Ref. [2] showed that lexical decision was facilitated by primes consisting of the target’s initial or final letters (see also [3]). Other evidence indicates that initial letters are more privileged than final letters in spatial coding. For instance, very brief exposures to two words can produce ‘migration errors’, where letters drift between words. Ref. [4] found that in such cases, only the first letters of each word tended to stay in place when letters were exchanged. Consequently, reading models such as the dual-route cascaded model [5] propose that first letters serve as *spatial anchors*, coding the positions of the remaining letters.

Acquired brain injury can produce various forms of dyslexia with distinct profiles. Surface dyslexia affects reading of irregular words [6,7,8,9]. Phonological dyslexia involves difficulty reading pronounceable nonwords [10,11,12,13]. Deep dyslexia is marked by semantic errors during reading [14,15]. In each case, these are termed *central dyslexias*, reflecting disruption to the central word-reading system [5]. Importantly, however, some reading deficits arise from attentional processing impairments peripheral to the central reading system. A cardinal example being the case of *attentional dyslexia* [16]. In this condition, patients often read whole words more accurately than they can name individual letters *within* those words. This dissociation is attributed to impaired attention: patients struggle to identify multiple items of the same type (e.g., letters) simultaneously, whereas single items (e.g., words) can be processed accurately [17]. Similar difficulties occur with multiple stimuli more generally (letters, numbers, words, or pictures), suggesting a category-specific perceptual-semantic overload due to faulty selective attention [18,19,20]. Put simply, in attentional dyslexia whole-word reading benefits from *coarse* multi-letter clusters or “word-form” templates that implicitly encode position information. These cluster-based codes can facilitate word recognition even when precise letter-by-letter coding is disrupted. Lexical knowledge can further guide the placement of these letter clusters into plausible positions within a word, compensating for individual letters that have migrated. Conversely, impaired attention reduces the resolution needed to track each letter’s exact location, leading to difficulties in recognising individual letters [17].

Importantly, although such coarse ‘top-down’ coding can support *word* reading, it will naturally fail for unfamiliar letter strings, consequentially impairing nonword reading. Attentional dyslexics often make visual errors on nonwords, such as letter transpositions that yield real words [21]. For instance, Ref. [22] described a patient (ED) who frequently read scrambled nonwords as familiar words (e.g., HMOE as “*home*”). They interpreted ED’s performance as using coarse positional coding of letter features that the lexical system partially corrects: the patient essentially tries to impose a plausible word onto the string; a classic example of the patient trying to ‘make lexical sense’ of the letter string [23]. Consequently, ED’s transposition errors were not random but guided by implicit knowledge of where letters are typically located in English words. Interestingly, this observation has also found that the tendency to ‘rearrange’ letters in nonword strings, to yield real words, almost never involves letter transposition errors that violate the first two letters in the string. That is there is a tendency, in letter-transposition errors (e.g., letters migrating between words), that initial letters tend to remain ‘fixed’ more often than internal letters. This echoes findings from healthy readers discussed earlier, such that this evidence suggests that first letters in a string serve as special anchoring points in orthographic processing and visual word recognition.

Inspired by this work, Ref. [1] examined the first-letter advantage in two attentional dyslexia cases (GK and FL; see [21]. Patients made lexical decisions on real words and on two types of nonwords: anagrams of those words (e.g., “radio”→“rdaio”) and control nonwords matched for frequency. Shalev and colleagues systematically varied the positions of transposed letters to test if patients were more sensitive to transpositions at *particular locations*. They predicted that patients would most easily reject nonwords with a transposition at the beginning of the word. Consistent with this prediction, nonwords with first-letter transpositions (like RCANE from CRANE) were rejected as often as control nonwords, despite other transposed-letter nonwords eliciting higher false positives. This pattern reinforces the idea that the first-letter position serves as a critical anchor in word recognition, even in disordered attention. Notably, patient GK had left visual neglect [21] yet *still* showed this first-letter advantage. This finding implies that binding of letter identity to the first position may occur *pre-attentively* or automatically. That is, the first-letter identity may be bound to its position even without *overt attention*, perhaps due to its special status in orthographic processing. Consequently, these findings challenge the assumption that all letter positions require equal attentional effort. Even with impaired attention, identity-location binding for the initial letter appears preserved. This supports theories positing automatic prioritisation of the initial position during word processing or a structural advantage due to its spatial separation from neighbouring letters. Whether from attentional weighting, coarse spatial codes, or lexical constraints, the first letter thus appears to act as a cognitive anchor point, central to both typical and impaired reading

Since many findings emphasise the importance of first letters, this begs the question of why this advantage manifests. One hypothesis is that letters at the start (or end) of a word have fewer neighbours and therefore suffer less attentional ‘load’ (i.e., lower *crowding*). Put simply, it may be that since the first letter sits ‘next’ to empty space, it is likely more visually salient. Ref. [24] tested this by removing regular spaces between words to equalise crowding and yet they still found a first-letter advantage. The authors interpreted these findings as indicating that the initial-letter advantage is probably not solely due to reduced interference but reflects how the reading system inherently encodes or uses the first letter. Other work has asked whether the first-letter advantage is *invariant* to stimulus orientation, or whether it is heavily dependent on a familiar *canonical* presentation. Ref. [25] found that transposed letter priming remains significant when English words are presented vertically (marquee style), suggesting that letters are recoded into an abstract ordinal framework (first, second, etc.) regardless of orientation. With this in mind, we tested whether the first-letter advantage described by [1] persists when words are presented *non-canonically*, specifically in a vertical (marquee) format.

It is worth mentioning that there are a variety of theoretical accounts for letter-position coding in word recognition, but they often suggest that the initial letter is particularly important. Some models incorporate *relative position* as part of their letter encoding schemes. In the Dual Route Cascaded model [5], for example, there is an “anchor point” against which subsequent letters are encoded. The first letter is the anchor, the second letter is encoded as first-letter +1, the third letter as first-letter +2 and so on. In this way, the initial letter is given an integral role in word recognition, and first-letter transposition is likely to be particularly disruptive. Ref. [26] SERIOL model considered letter-position coding as being *temporal* rather than spatial in nature—in other words, the initial letter is encoded as such because it is activated before the second letter in time. This is important for our study, in that the initial letter of a string is activated first even if the stimulus is presented in marquee format.

Our study thus addresses two questions: (a) can we replicate the first-letter advantage in two cases of posterior cortical atrophy (PCA)—a neurodegenerative syndrome characterised by severe attentional deficits and (b) in order to probe whether this first-letter advantage is *orientation invariant*, we ask does this advantage persist when stimuli are presented in a non-canonical (vertical) format? Recent work has demonstrated that changing the presentation format for written words in reading can have consequences on PCA performance consistent with the fact that such formats increase attentional load [27]. Importantly, since PCA patients often show extreme crowding and simultanagnosia (inability to perceive multiple items simultaneously), they are an ideal context to study attentional effects of visual word recognition and reading. Notably, Ref. [28] found that in two PCA cases initial letters were almost always preserved in reading errors (e.g., 93% preservation vs. 74% for medial letters in one patient). These patients made far more mistakes on middle letters than on first letters of words, suggesting again that the first-position letter is less vulnerable under extreme impairment. To date, no study has examined PCA patients with visual word recognition tasks that manipulate nonword letter arrangements or presentation format. We therefore investigate whether the ‘privileged’ first-letter effect appears in PCA cases and to what extent it remains robust to changes in format.

## 2. Case Descriptions

Posterior cortical atrophy (PCA) is typically diagnosed on the basis of insidious onset and gradual progression of symptoms, with early, prominent disturbance of visual or posterior cognitive functions and relative sparing of language, memory and executive abilities in the initial stages. As a consequence, early diagnostic criteria can include cognitive deficits such as visuospatial disorientation, simultanagnosia, object or face recognition problems, constructional and dressing apraxia, and alexia or acalculia. Supportive features are posterior cortical atrophy on MRI or hypometabolism on PET. Other exclusion criteria can include alternative structural or ophthalmological causes and presentations dominated by memory or language decline. In both our cases initial diagnosis was provided by a neurologist on the basis of the pattern of features described above, which included neurological, radiological and behavioural assessment by a older adult psychiatrist and a specialist neuropsychologist, at relevant hospital clinics.

KL, a 62-year-old right-handed man with 16 years of education, experienced 4 years of progressive visual difficulties impacting his work as a computer programmer. He struggled to read code due to characters “*moving around on the screen*,” and reported slower thought processes and planning difficulties, though he denied issues with language or episodic memory. A SPECT scan showed reduced perfusion in the parietal and posterior temporal lobes, and a brain MRI revealed occipitoparietal/occipitotemporal atrophy with widened parieto-occipital sulci, consistent with posterior cortical atrophy (PCA) [29]. KL was featured in a case-series on mental imagery in PCA [30], confirming severe imagery impairments and attention-based object/visuospatial deficits, with spared language and memory (full neuropsychological details in that paper). His reading skills were assessed with PALPA tests: excellent word reading (60/60 regularity, 80/80 frequency/imageability), mild nonword reading impairment (21/24, suggesting phonological dyslexia linked to attention issues; cf., Refs. [31,32], and strong performance in lexical decision (120/120), rhyme judgement (55/60), letter naming/matching (26/26 each), and spelling (35/40 oral, 35/40 written). Overall, KL exhibits PCA with attention deficits but no significant language or word reading issues at this stage.

GS is a 57-year-old right-handed woman with 11 years of formal education. Over the past 4 years, she has experienced worsening visual difficulties that have significantly affected her role as a care worker. She struggles to locate objects directly in her field of vision and has poor depth perception, causing her to feel anxious about lifting patients in and out of bed. She also finds it challenging to recognise familiar people from a distance and can no longer read cursive writing, though she can manage printed text with effort, describing the letters as *“floating around”* and *“sometimes not aligning properly.”* She has experienced incidents of mistaken identity, such as shouting at a sheep thinking it was her dog, only realising her error as her dog approached. She reports no issues with language expression or comprehension, and her memory for both recent and past events remain reasonably intact. However, she experiences mild anxiety and depression due to her inability to work or drive. Like KL, GS was featured in a case series on mental imagery in PCA [30], confirming severe imagery impairments and attention-based object/visuospatial deficits, with spared language and memory (see paper for full testing details). As a comparator to the tests discussed for KL; GS showed relatively preserved word reading (58/60 regularity, 74/80 frequency/imageability), but severe nonword reading impairment (12/24), and interestingly inspection of her errors indicated that 9/12 were real word responses (e.g., Birl—“Bird”, Bem—“Beam”, Squate—“Squat”), but in all cases *first two letters* were preserved. Again, GS’s performance indicates that phonological dyslexic like behaviour can manifest in the context of attention issues. Other testing also revealed emergent mild impairments with lexical decision (104/120—most errors false positives), rhyme judgement (41/60) and spelling (19/24 oral, 18/24 written); though single letter naming/matching (26/26 each) was entirely intact. Overall, GS presents with clear attentional deficits and emerging deficits in some language-based tasks, consistent with the principle that she is likely more progressed in neurodegenerative profile to that of KL. All aspects of our study were conducted in accordance with the Declaration of Helsinki and approved by an IRAS Ethics committee (Project ID: 234933 and date: 22 October 2019) and informed consent was obtained from all participants involved in the study.

## 3. Experiment 1—Replicating the Study of Shalev and Colleagues

### 3.1. Methods

All test materials used in this study were a direct replication of the items taken from the study of Shalev and colleagues with their two attentional dyslexia patients [1], as a consequence to prevent redundant duplication we would suggest readers refer to their work for the construction and selection of the key stimulus sets. Following the work of Shalev and colleagues both patients (KL/GS) undertook a series of lexical decision task experiments—in each case the same 60 five letter word targets were used and five letter nonword foils comprised two types—(a) transposition nonword foils, which comprised a word where either the first two letters (1 + 2, “owner”→“woner”), second/third letters (2 + 3, “owner”→“onwer”), third/fourth letters (3 + 4, “owner”→“owenr”), or last two letters (4 + 5, “owner”→“ownre”), were transposed or (b) control nonword foils which were novel (but pronounceable) letter strings. For each of these experimental nonwords, there was a control non-word formed by substituting two different letters to the transposed letters, matching the letters for visual similarity (ascenders substituting for ascenders, descenders for descenders, etc.—e.g., ‘woner’→‘vaner’; ‘onwer’→‘omver’; ‘owenr’→‘owumr’; ‘ownre’→‘ownsu’). All items were provided from the original study undertaken by Shalev and colleagues, and our procedure followed theirs as closely as possible (see Experiment 2—[1]). These five lexical decision studies were administered across five testing sessions, a week apart to avoid practice effects. All test materials were administered via experimental software presentation (Psychopy: version 3.0.1) on a laptop at the participant’s homes, and this software computed accuracy for all items presented. In all cases participants were instructed to respond orally whether they thought the letter string formed a word or nonword (at their own pace as accurately as possible), while the experimenter pressed the corresponding key on the keyboard (‘L’ for word and ‘A’ for nonword). All letter strings were presented lower case and 80-point font size. Thus, accuracy was our key dependent variable. Case KL was tested on two occasions with an 18-month duration between these occasions, this was in order to determine whether his pattern of performance might change as his neurodegeneration progressed, and whether the observed patterns were replicated. A different order of the testing conditions was used in each case. We had intended to do the same with GS, but initial testing determined she was already generally much poorer at the task than KL, and additionally the outbreak of a global pandemic (COVID-19) prevented the practicality of a subsequent patient home visit.

### 3.2. Analysis

To quantify lexical decision performance across experimental conditions, we computed signal detection theory (SDT) metrics for each condition using R (version 4.4.1). Specifically, we calculated sensitivity (d′) and response bias (c) based on participants’ ability to distinguish real word targets from nonword foils. Hits were defined as correct “word” responses to real word targets, and false alarms as incorrect “word” responses to nonword foils of the relevant type. Sensitivity (d′) was computed as the difference between the z-transformed hit rate and false alarm rate, and response bias (c) as the negative half-sum of these z-scores. To provide uncertainty estimates, 95% confidence intervals for both d′ and c were calculated using a nonparametric bootstrap procedure implemented in R’s boot package. This involved resampling the trial-level data with replacement (1000 iterations) within each foil type combination. This approach provides robust, distribution-free confidence intervals that do not rely on binomial or normal approximations and is well-suited to experimental datasets with unbalanced or limited trial counts. The full analysis pipeline, including data processing and bootstrapped SDT estimation, was implemented in R and is available upon request.

To investigate the effect of condition on binary response accuracy, a binomial logistic regression model was fitted. Data were analysed in RStudio 2025.05.0 using the *tidyverse* and *readxl* packages for data wrangling and import—the variable *condition* was treated as a categorical predictor by converting it to a factor (i.e., conditions 1 through 4—see Table 1). A generalised linear model (GLM) with a binomial family and logit link function was used to predict the probability of a correct response (*Response. corr*) as a function of *condition*. The model took the following form:


*glm (Response. corr ~ condition, family = binomial, data = data)*


Model outputs were summarised to assess overall condition effects. To further explore pairwise differences between conditions, estimated marginal means and pairwise comparisons were conducted using the *emmeans* package with results reported on the response (probability) scale. All data and R codes are available on request.

### 3.3. Results

#### 3.3.1. Case KL—Pattern of Observations

KL:

Table 2 presents KL’s sensitivity (d′) and response bias (c) for transposed and control nonword foils across the four lexical decision versions (conditions) in our first testing period. In condition 1, where nonwords involved transpositions of the first two letters (see Table 1; condition 1 = 1 + 2), sensitivity was *equally* high for both transposed and control nonwords (d′ = 4.54 for both, with accuracy being *100%*), suggesting no observable performance cost for this type of letter displacement. In conditions 2–4, a consistent pattern emerged: sensitivity was markedly lower for transposed foils compared to control nonwords. Observations of response bias (c) further suggested a general pattern of near zero or even slightly conservative bias for control foils; whereas transposed foils consistently produced negative c values, indicating a liberal response bias (i.e., a greater tendency to misclassify transposed nonwords as real words), with again this bias being much lower for 1 + 2 nonword foils.

Table 3 again presents KL’s sensitivity (d′) and response bias (c) for transposed and control nonword foils across the four lexical decision versions in our second re-testing occasion, 18 months later. It is apparent that consistent with his neurogenerative presentation, performance has generally declined somewhat. However, the key pattern observed in the previous occasion remains. That is for nonwords that involved transpositions of the first two letters (1 + 2), sensitivity is much higher than for all the other transposition nonword formats, and much closer to that of the matched control nonwords. In all cases there is no overlap in d’ confidence intervals for 1 + 2 nonwords and any other transposed nonword foils. Observations of response bias (c) again were consistent with a general pattern of near zero or even slightly conservative bias for control foils; whereas transposed foils (other than 1 + 2) consistently produced negative c values, indicating a continuing pattern of liberal response bias. In sum, despite an overall decline in performance, KL continued to show a selective preservation of sensitivity when rejecting transposed-letter nonwords in the 1 + 2 condition. In contrast, later transpositions (2 + 3, 3 + 4, 4 + 5) produced substantial decrements in sensitivity and stronger liberal response bias, indicating increased lexical interference and uncertainty. Acc = Accuracy

Since in the second testing period accuracy was not perfect for the critical initial transposition condition, we sought to statistically evaluate whether accuracy differed across the four transposition conditions in KL’s second testing occasion under canonical (horizontal) word presentation. A logistic regression model was used to assess the effect of occasion (test condition) on response accuracy. To do so, we conducted a logistic regression analysis with trial-level accuracy (0 = incorrect, 1 = correct) as the outcome variable and condition (1 = 1 + 2 transposition, 2 = 2 + 3, 3 = 3 + 4, 4 = 4 + 5) as a categorical predictor. Condition 1 served as the reference level, allowing direct comparison of later transposition positions against the initial two letters, and thus provided a formal statistical test of whether that advantage generalised across all other positional contrasts. The model revealed a significant main effect of condition, with performance differing substantially across the four conditions. Relative to condition 1, accuracy was significantly lower in 2 (β = −3.04, z-score = −3.71, *p* = 0.0002), 3 (β = −4.03, z-score = −4.67, *p* < 0.0001), and 4 (β = −2.91, z-score = −3.55, *p* = 0.0004). Estimated marginal means indicated a high probability of a correct response in condition 1 (93.3%), which dropped to 40.0% in 2, 20.0% in 3, and 43.3% in condition 4. These results indicate a marked drop in accuracy relative to the first letter transposition type, with subsequent conditions showing similarly reduced performance levels. These results strongly support the hypothesis that transpositions affecting early letter positions are easier for the patient to detect and reject, consistent with privileged position encoding at the beginning of letter strings. KL’s findings, replicated across testing periods, are consistent with the work of Shalev and colleagues and indicate that early-position transpositions exert minimal disruption on lexical decision, consistent with position-sensitive orthographic encoding mechanisms.

#### 3.3.2. Case GS—Pattern of Observations

Table 4 presents d′ and c values for Patient GS across the four versions of lexical decision experiments. As with KL, of all the transposition nonword types, GS performs best with those that are first two letter transpositions (1 + 2), with markedly reduced sensitivity for other types. Confidence intervals showed only minimal or no overlap, suggesting reliable deficits in discriminating word-like transpositions. In addition, generally such nonwords provoked a more liberal response bias (like KL) with a greater tendency to misclassify such transposed nonwords as real words (false positives). To formally assess whether the patient’s accuracy differed across the four types of transposed foil condition, we again conducted a binomial logistic regression analysis approach with trial-level accuracy (0 = incorrect, 1 = correct) as the dependent variable and condition (coded categorically) as the predictor. The model revealed a significant overall effect of condition on accuracy (*χ*^2^(3) = 14.43, *p* < 0.01). Compared to condition 1, accuracy significantly decreased in 2 (β = −1.15, *z* = −2.08, *p* = 0.038) and 3 (β = −1.86, *z* = −3.24, *p* = 0.001), while condition 4 did not differ significantly from condition 1 with equivalent levels of accuracy (see Table 3).

Overall, a consistent pattern emerged across both patients, which indicates a *privileged status* for first/second letter positions in lexical processing, such that both cases showed highest levels of sensitivity (d′) for transposition nonwords of that type. These findings align with theories of visual word recognition that emphasise the functional importance of word-initial letters for lexical access in visual recognition, such that disruptions to early letter positions are more easily detected, while transpositions in less salient positions are much more likely to be compromised.

## 4. Experiment 2—Non-Canonical Presentation Format

Having established that *both* our cases presented with a pattern of performance consistent with that reported in the work of [1]; namely higher accuracy at rejecting nonwords with first two letter transpositions—confirmed through both more rigorous analysis than the original study and replication for KL. We were interested in exploring whether a similar pattern might be observed with letter strings presented in *non-canonical* formats. A key example of such a non-canonical format is the presentation of items in a vertical or marquee manner (i.e., with letters presented north–south akin to certain street signage)—very recent work undertaken by the authors has explored this format change in PCA and found it can disrupt reading aloud [27]. As a consequence, we wanted to explore this further in this case. Put simply, does the first two letter advantage generalise to letter strings presented in a vertical (marquee) presentation format? This would enable us to determine whether the key two letter advantage is *specific* to the typically familiar word form orientation or not. In this case, the experiment was only undertaken with KL; given GS was generally poorer at performing in experiment 1, she was reluctant to participate in this experiment.

### 4.1. Methods—KL Experiment 2—Non-Canonical Format Presentation

In this case, the experiment was largely identical to that in the previous study, with two key exceptions. Firstly, for the critical five letter nonword foils in the case of transposition nonword foils, we employed only two conditions: first two letters (1 + 2, “owner”→“woner”), and last two letters (4 + 5, “owner”→“ownre”) with their matched control nonword foils. Since we found no evidence that other conditions differed for KL, we felt it best to simplify the experiment and save time for the participant. Secondly, all items for a given experiment were presented vertically (marquee) with letters in a north–south configuration. These two lexical decision studies were administered a week apart to avoid practice effects. As before, all test materials were administered via experimental software presentation on a laptop at the participant’s home and all other aspects of the procedure were identical to experiment 1. In addition, as in experiment 1 KL was tested on two occasions with an 18-month duration between occasions, this was to determine whether his pattern of performance might change as his neurodegeneration progressed, and whether the observed patterns were replicated. A different order of the testing conditions was used in each case. To reiterate, this study sought to explore whether the first letter transposition nonword advantage remained even when items were presented either in a vertical (marquee) format and confirmed in a second testing occasion.

### 4.2. Results—KL for Non-Canonical Presentation Formats—Experiment 2

In Table 5 we present the sensitivity (d’) and bias (c) for KL on the two kinds of nonword transposition conditions (1 + 2 and 4 + 5) in both initial and replicated testing periods. What is apparent is that in both testing periods sensitivity (d′) remained higher in the 1 + 2 transposition condition relative to the 4 + 5 transposition condition, with either minimal (period 1) or no (period 2) overlap in confidence intervals. Similarly, comparative sensitivity performance for 1 + 2 and 4 + 5 on transposed nonwords with nonword controls suggests near equivalent accuracy for the former. The pattern of response bias (c) further supports this asymmetry. While c remained positive or near neutral for control and transposed foils in the 1 + 2 condition, indicating a cautious decision strategy, it shifted markedly in the 4 + 5 transposition condition. Specifically, c became negative for transposed foils, reflecting a liberal bias and greater tendency to misclassify these items as words that mimics the pattern seen earlier with canonically presented items.

For testing time period 1, to determine whether KL’s previously observed advantage for early-letter transpositions extended to vertical word presentation, we conducted a logistic regression analysis using trial-level accuracy (0 = incorrect, 1 = correct) as the dependent variable and transposition condition (1 + 2 vs. 4 + 5) as a categorical predictor. Condition 1 (1 + 2 transpositions) served as the reference level. The model revealed a significant effect of condition, with performance declining in condition 4 compared to condition 1 (β = −1.93, *p* = 0.0067). Estimated marginal means indicated that the probability of a correct response was high in condition 1 was 90% but dropped to 57% in condition 4. In addition, the analysis revealed a significant decline in performance from condition 1 to condition 4 (β = −2.37, *p* = 0.0038). Estimated marginal means showed again a high probability of correct response in condition 1 of 93% which decreased to 56% in condition 4. The pattern of results across both time periods supports the hypothesis that initial letter positions are more robustly encoded, even in the context of attentional reading impairments and a non-canonical (vertical) presentation format. Overall, these results mirror KL’s performance in the canonical format, reinforcing the privileged status of early letter positions in word recognition, even when visual presentation is *nonstandard*.

## 5. General Discussion

### 5.1. Replication of the First-Letter Advantage in Canonical Presentation

Our results have demonstrated a clear *first-letter* advantage in orthographic processing in visual word recognition for both PCA patients. There was a consistent pattern that nonwords created by transposing the first *two letters* of a word were rejected with higher accuracy than nonwords formed by transpositions in later letter positions. Notably, no such gradient in performance emerged across the four other control nonword conditions. Our work thus replicates the key finding of [1], now extended to patients with PCA, and underscores that this first-letter advantage persists in a population with known spatial attention deficits. Even under these constraints of disrupted attention, the initial letter of a word appears to serve as a *cognitive anchor* for the encoding of orthographic input in lexical decision.

### 5.2. Generalisation to Non-Canonical Word Formats

Crucially, we extended this line of inquiry by exploring whether the same advantage held when words were presented in a *non-canonical* spatial orientation. Specifically, we tested the key effect using vertically presented stimuli (marquee format), across two testing periods to enable replication. Again, we observed a clear pattern: the 1 + 2 transposition condition consistently yielded better accuracy and d’ sensitivity than a 4 + 5 transposition condition, with statistical significance confirmed using logistic regression. These results support the hypothesis that the first-letter advantage is not simply a result of its location on the *leftmost* part of the visual field (in canonical horizontal text) but reflects a more abstract positional status tied to its identity as the ‘initial’ character in a word sequence. This interpretation is consistent with key findings from *neglect dyslexia*; such patients (often with right-hemisphere damage) consistently fail to report or misread one side of words—typically the left side—for instance reading the word *snowball* as “ball” missing the first few letters. However, work with such cases has revealed that the coding of letter positions is clearly linked to the relationship between spatial attention and how the brain represents coordinates of the word. In a now classic study, Ref. [33] demonstrated that letter positions are encoded in a word-based reference frame rather than purely by retinal coordinates. They reported a neglect patient who would typically drop letters from one end of a word, but this depended on the *canonical orientation* of the word. For example, when words were presented vertically or even mirror-reversed, the patient’s errors still involved the *initial letters* of the word in reading order, not simply the leftmost letters on the page. The authors interpreted this as suggesting that the visual word recognition system clearly codes which letter is ‘first’ in the word’s *particular* orientation, such that the neglect then appears to reflect that *portion* of the word, despite the actual physical orientation. Put simply, neglect dyslexia errors are *word centred* rather than an issue with a particular spatial location in visual space. Such findings imply that once the visual system identifies the word stimulus, it maps it onto an abstract representation where letter positions are defined relative to the word’s ‘start’ and ‘end’. The first letter is ‘first’ even if it is not on the typical extreme left in viewer coordinates, and thus it becomes the target (or victim) of neglect based on that abstract positioning. In summary, neglect cases reinforce two points: (1) letter-position coding is abstract and tied to the concept of *beginning* of a word, and (2) adequate spatial attention is necessary to utilise that information. When attention to one side is lost, even a privileged first letter can be omitted entirely, explaining the pattern of left-initial letter omissions in neglect dyslexia. Put simply, the brain encodes ‘first letter’ status in an *orientation-invariant* way. The first letter is not just another letter; it is marked by being at the start of the sequence, and that status survives transformations like inversion or vertical arrangement. Our findings echo this logic: the first letter remains a structurally marked entity across different visual configurations, consistent with orientation-invariant coding of word-initial letters. Moreover, our findings also challenge *position-specific* slot models of reading [34] and instead favour frameworks where encoding is positionally graded, flexible, and influenced by lexical and structural knowledge (e.g., dual-route cascaded models, open-bigram coding).

### 5.3. Theoretical Implications for Models of Letter-Position Coding

The results from both canonical and non-canonical formats bolster models in which letter identity and letter position are at least partially dissociable. Open-bigram models, for example, propose that letter order is coded via overlapping bigrams; thus the word *TRAP* might be represented by the set {T-R, T-A, T-P, R-A, R-P, A-P}, implicitly encoding *relative order* without pinning each letter to an exact slot. Such a scheme naturally gives a stronger role to the first letter, since the first letter participates in many of the bigrams (combining with every letter that follows), whereas a middle letter only pairs with letters after it. This could explain why the first letter’s identity is so influential—it reverberates through multiple element pairs in the internal code. Similarly, the dual-route cascaded model [5] and the MORSEL model [35] both emphasise the special role of the first letter in early orthographic encoding. In these accounts, the first letter is not just another element in the sequence, but an *anchor* that defines the spatial scaffold for decoding the rest of the string. In the context of attentional dyslexia and PCA, where deficits in spatial attention can lead to transpositions and crowding errors, our findings suggest that initial-letter encoding is robust enough to remain intact. This has implications for how spatial attention interacts with orthographic processing: although patients may struggle with encoding the absolute locations of internal or final letters, their performance on transposed nonwords implies that the system still recognises the importance of beginning-letter structure. This could reflect either automatic attentional weighting of the first letter, or perhaps a structural advantage tied to its unique positioning at the boundary of the string. Indeed, this resonates with data from [28], who reported that initial letters were significantly more likely to be preserved than medial letters in PCA patients. The present findings converge with this interpretation: the edge letters, particularly the first, are less vulnerable under conditions of attentional impairment.

## 6. Limitations

While the present findings provide novel insights into letter-position coding in attentional dyslexia and PCA, several limitations must be acknowledged. First and foremost, the study is based on two detailed case studies. Although this design allows for in-depth analysis of individual performance patterns and provides important observations of how reading can be specifically disrupted in PCA, in a manner akin to that reported for other patients, the small sample necessarily limits the generalisability of the findings. As a consequence, our analysis has been to focus on the *relative advantage* of letters in a particular spatial position of a word. A larger sample would have provided a more obvious comparison to a comparative healthy control sample, a point made by one our reviewers. We would of course agree, but our results are to provide an important early illustrative observation of the very real struggle such patients can face, which can motivate further larger scale work.

A second challenge relates to using a logistic regression approach to estimate the degree to which the relative first two letter advantage observed in each instance with a particular case was statistically significant, to complement the observations in the bootstrap approach. A reviewer has made the point that such an approach assumes that observed trials are independent and so can inflate SE values with consequential misinterpretation of model estimates. We would agree, though again our attempt here was aimed to be illustrative and the findings are entirely consistent with the patterns observed from the bootstrap approach we also utilised. Nonetheless, we undertook subsequent investigations of any likely impact of such dispersion effects and found them likely to be quite minimal, whilst model comparison approaches (using quasibinomial specification) indicated again that dispersion impacts were not serious. However, we would stress that caution should be taken in the interpretation of these illustrative regression models.

Thirdly, questions remain regarding the extent to which these effects would generalise *beyond* English, since although they impact models and theories based on this language specifically, it is likely other languages pose different challenges/have advantages [36]. A reviewer quite rightly pointed out that writing systems vary considerably in their orthographic and positional constraints. For example, in Semitic languages letters change form depending on their position within a word, which reduces the likelihood of simple transposition errors; in German, the capitalisation of all nouns introduces a confound in the salience of first letters. These differences raise the possibility that the “first-letter advantage” observed here reflects language-specific properties of English orthography rather than a universal positional status. We would certainly agree, and thus future work across different languages and orthographic systems would be helpful to clarify this issue, particularly as might pertain to the ‘risks’ of specific reading problems manifesting in PCA populations of particular native languages.

Finally, as a reviewer pointed out, the study did not attempt to model individual learning dynamics or adaptive lexical processing, as has been proposed in recent computational and educational frameworks (e.g., adaptive recommendation systems for dyslexic learners). While such approaches fall outside the scope of this study, their absence represents an additional limitation in quantifying subject-specific variability. Together, these considerations underscore the importance of treating the current findings as case-based demonstrations that inform theoretical accounts of orthographic processing, while exercising caution in making broader generalisations.

## 7. Conclusions

Taken together, our study replicates and extends a foundational finding in the reading literature: that the first letter in a word is perceptually and cognitively privileged. This privilege appears robust even in the face of severe attentional deficits and presentation distortions. It offers strong support for word-centred, orientation-invariant representations of letter position and reinforces the idea that the beginning of a word serves as a foundational spatial cue in the visual recognition process. Beyond reinforcing the importance of initial-letter anchoring, these findings open several avenues for future work. Could training paradigms leverage this robustness to support reading interventions in PCA or other neurodegenerative disorders? As we have mentioned earlier, would these effects generalise to languages with different orthographic directions? Future research may also explore whether similar anchoring effects are observed in oral and written spelling to determine the degree of generalisation of first letter lexical privilege.

In conclusion, the present work provides compelling evidence that the first letter enjoys a privileged role in visual word recognition, even under conditions of impaired attention and atypical presentation format. This has implications for models of orthographic processing, clinical diagnosis, and potentially, therapeutic intervention. Put simply, *where* a letter is in a string influences *what* word we see.

## Figures and Tables

**Table 1 brainsci-15-00967-t001:** Examples of four nonword foil types (base word = owner).

Nonword Presentation Condition	Nonword Transposition Foils	Nonword Control Foils
1: First two letters (1 + 2)	woner	vaner
2: 2nd/3rd letters (2 + 3)	onwer	omver
3: 3rd/4th letters (3 + 4)	owenr	owumr
4: Last two letters (4 + 5)	ownre	ownsu

**Table 2 brainsci-15-00967-t002:** KL performance across presentation formats and foil types.

KL—Canonical—Time 1	Foil Type	Acc	d′	95% CI (d′)	c	95% CI (c)
Letter transposition (1 + 2)	Control	1	4.54	[4.64, 4.58]	−0.13	[−021, 0.05]
Letter transposition (2 + 3)	Control	1	4.11	[3.65, 4.58]	0.09	[−0.20, 0.31]
Letter transposition (3 + 4)	Control	1	3.72	[3.28, 4.54]	0.28	[−0.13, 0.49]
Letter transposition (4 + 5)	Control	1	3.72	[3.28, 4.54]	0.28	[−0.13, 0.49]
Letter transposition (1 + 2)	Transposed	1	4.54	[4.45, 4.58]	−0.13	[−0.22, 0.05]
Letter transposition (2 + 3)	Transposed	0.6	2.21	[1.58, 3.02]	−0.86	[−1.25, −0.54]
Letter transposition (3 + 4)	Transposed	0.73	2.18	[1.59, 3.06]	−0.49	[−0.88, −0.15]
Letter transposition (4 + 5)	Transposed	0.87	2.63	[1.99, 3.67]	−0.26	[−0.68, 0.12]

**Table 3 brainsci-15-00967-t003:** KL performance across presentation formats and foil types on retesting.

KL—Canonical—Time 2	Foil Type	Acc	d′	95% CI (d′)	c	95% CI (c)
Letter transposition (1 + 2)	Control	1	3.72	[3.31, 4.55]	0.28	[−0.13, 0.48]
Letter transposition (2 + 3)	Control	0.93	2.74	[2.13, 3.68]	0.03	[−0.35, 0.50]
Letter transposition (3 + 4)	Control	1	4.11	[3.68, 4.58]	0.09	[−0.19, 0.30]
Letter transposition (4 + 5)	Control	0.97	3.63	[2.95, 4.56]	−0.15	[−0.58, 0.25]
Letter transposition (1 + 2)	Transposed	0.93	2.98	[2.37, 4.11]	−0.09	[−0.51, 0.36]
Letter transposition (2 + 3)	Transposed	0.40	1.09	[0.46, 1.80]	−0.79	[−1.14, −0.50]
Letter transposition (3 + 4)	Transposed	0.20	1.16	[0.40, 1.95]	−1.39	[−1.82, −1.07]
Letter transposition (4 + 5)	Transposed	0.43	1.80	[1.18, 2.31]	−1.06	[−1.46, −0.77]

**Table 4 brainsci-15-00967-t004:** GS performance across presentation formats and foil types.

GS—Canonical	Foil Type	Acc	d′	95% CI (d′)	c	95% CI (c)
Letter transposition (1 + 2)	Control	0.87	2.4	[1.82, 3.29]	−0.14	[−0.47, 0.26]
Letter transposition (2 + 3)	Control	0.87	1.67	[1.11 2.49]	0.22	[−0.10, 0.64]
Letter transposition (3 + 4)	Control	0.80	1.75	[1.18, 2.50]	−0.07	[−0.39, 0.29]
Letter transposition (4 + 5)	Control	0.97	2.32	[1.74, 3.09]	0.5	[0.19, 0.86]
Letter transposition (1 + 2)	Transposed	0.73	1.94	[1.36, 2.75]	−0.37	[−0.76, −0.06]
Letter transposition (2 + 3)	Transposed	0.47	0.53	[−0.04, 1.08]	−0.35	[−0.63, −0.09]
Letter transposition (3 + 4)	Transposed	0.30	0.44	[−0.14, 1.03]	−0.73	[−1.05, −0.44]
Letter transposition (4 + 5)	Transposed	0.73	1.26	[0.75, 1.93]	−0.03	[−0.30, 0.30]

**Table 5 brainsci-15-00967-t005:** KL performance across vertical presentation formats and foil types.

**KL—Vertical—Time 1**	**Foil Type**	**Acc**	**d′**	**95% CI (d′)**	**c**	**95% CI (c)**
Letter transposition (1 + 2)	Control	0.97	2.67	[2.10, 3.56]	0.32	[−0.01, 0.67]
Letter transposition (4 + 5)	Control	0.90	2.37	[1.82, 3.39]	0.03	[−0.34, 0.46]
Letter transposition (1 + 2)	Transposed	0.90	2.22	[1.63, 3.18]	0.10	[−0.22, 0.57]
Letter transposition (4 + 5)	Transposed	0.57	1.32	[0.77, 1.99]	−0.50	[−0.82, −0.21]
**KL—Vertical—Time 2**	**Foil Type**	**Acc**	**d′**	**95% CI (d′)**	**c**	**95% CI (c)**
Letter transposition (1 + 2)	Control	0.93	1.92	[1.28, 2.80]	0.44	[0.13, 0.88]
Letter transposition (1 + 2)	Control	0.87	1.53	[0.98, 2.43]	0.29	[0.02, 0.73]
Letter transposition (4 + 5)	Transposed	0.90	1.92	[1.28, 2.80]	0.44	[0.13, 0.88]
Letter transposition (4 + 5)	Transposed	0.57	0.63	[0.06, 1.19]	−0.15	[−0.45, 0.13]

## Data Availability

The original data presented in the study are openly available in Open 500 Science Framework at https://osf.io/f7che/ (accessed on 4 September 2025).

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
