# Peer review of "The First Shall Be First: Letter-Position Coding and Spatial Invariance in Two Cases of Attentional Dyslexia"

_brainsci, 2025, doi:10.3390/brainsci15090967_

Round 1

Reviewer 1 Report

Comments and Suggestions for Authors

This study is based on case studies of two individuals with posterior cortical atrophy (PCA). The study reports the performance of these two individuals in lexical decision tasks which included real words, nonword foils constructed by transposing letters in real words, and control nonwords.

Generally, I found the topic to be interesting and the manuscript to be well-written. As my comments below reflect, a key issue with the current version of the manuscript concerns the methods and results. I would strongly advise the authors to pay more attention to these sections, which should lead to overall improvement. Please see all my comments below.

1) Pages 5 and 6 (lines 201-209)

The description of the (non)word stimuli led to confusion. It took me a few re-reads till I understood that for each of the type (a) transposition nonword foils, there was a control nonword foil (type (b)) which was a letter string containing letters that were visually similar to the transposed letters in the type (a) items. At least, this is what I understood, but I am still not sure, Actually, the tables with results on subsequent pages helped clarify the manipulations. I would recommend adding a table to present the experimental items with examples on page 5/6.

2) Page 6 (lines 213)

It is good practice to specify what software (and version) is used in experiments.

3) Page 6 (lines 232-233)

The authors write: “To mitigate ceiling and floor effects, we applied a log-linear correction to both hit and false alarm rates [32]”. I do not understand how merely transforming the data can “mitigate ceiling and floor effects”, beyond just compressing the data to reduce skew which is the most common use of the log transformation as far as I am aware. Could the authors explain further?

4) Page 6 (lines 243-250)

The paragraph where the modelling approach is explained led to confusion. Firstly, the paragraph starts by mentioning “To investigate the effect of session…”. For one thing, differences between the five testing sessions (line 211-212) are not linked to the hypotheses the authors formulated, so it comes a bit out of the blue for the reader. For another thing, the authors do not actually include session as a predictor in their formula later on (line 250). Instead, they have included condition, which I guess stands for the letter transposition manipulation (i.e. 1+2, 2+3, 3+4, 4+5). This is what I could infer based on the Table results. The Table results also suggest that another important factor to include in models is Foil Type. Yet, this is also not included in models, much like Session. Finally, a related issue emerges later on page 8 (lines 290-291) where the authors write “...the effect of session (test condition)…”. This causes even more confusion as to what the terms “session” and “condition” actually stand for, and why the authors chose to use these terms interchangeably (see also lines 320-321), if they refer to different experimental aspects, as one would expect. The authors should clarify the above to resolve confusion.

5) Page 6 (line 250)

The choice of a generalised liner model for the type of data presently reported is inappropriate. Specifically, the glm model violates the statistical assumption of independence of observations as the authors have repeated measurements from the same participants. A glmer, for example, would avoid this problem if the authors decided to analyse both the KL and GS data together, and have “participant” as a random effect. Yet, in practice, such a model would again fail most likely, as between participant variance cannot be effectively estimated with such a small sample size (N=2). In my experience, when dealing with case studies, statistical analyses are not practically feasible, and even if one manages to get the models to run, SE values would be large enough to cast doubt on the reliability/precision of model estimates. A more common approach when dealing with case studies is to report degree of deviation from normative values, either using assessments standardised based on the performance of healthy controls or by recruiting an age-matched control group as a reference point. Given that the study presently reported does not include the above, I can’t think of other statistical analysis suggestions I could propose. Thus, I would advise the authors to focus on descriptive stats (i.e. central tendencies such as mean accuracy) and comment on the absence of statistical analyses and normative values in a Limitations section.

6) Tables 1-4

The authors could include another column in the Tables to report on mean accuracy (e.g. in %). This information is currently embedded in the text or is not reported (e.g. line 287 “accuracy was not perfect”). It would be better to have all relevant performance measures presented together in each Table.

7) Page 8 (lines 296-301)

The following comment may become irrelevant if the authors remove the statistical analyses as I advised them in my comment 5). If that is the case, please ignore the below.

In lines 296-301, the authors report statistical analyses results for the performance of KL in the second testing period. Yet, they do not do the same on the previous page (page 7) for the first testing period. The authors should report both stat results and if there is any inconsistency in outcomes, they should comment on this in the Discussion section.

Additionally, in their supplementary materials uploaded on OSF (https://osf.io/f7che/) the authors have included their R code for the results reported on page 10 (lines 383-389), but not for the ones reported on Page 8 (lines 296-301) or for the other statistical results reported in the case of GS (lines 318-328). Please upload all scripts for replicability.

8) Page 9 (lines 324-328)

The two sentences in lines 324-328 essentially repeat the same information. I would remove one of them.

9) Page 12 (lines 483-484)

Finally, I found the introductory section and the discussion to be well-researched and well-written. The subsection “Implications and Future Directions” included the question “Would these effects generalise to languages with different orthographic directions?”. This is a key question, but I think there are more outstanding questions to address that should be mentioned. For one thing, the lack of a healthy control sample to which the performance of KL and GS can be compared is a limitation of this study, and should be addressed in a Limitations section, as mentioned in my comment 5). For another thing, beyond different orthographic directions in other languages, there is also other variability across writing systems that, in my view at least, raise questions regarding the “universality” of the first-letter advantage reflecting “a more abstract positional status” (line 416). While the attribution of the observed results to positional status may hold for English, it is unclear to me that the same could be said about typologically distant languages. For instance, in Semitic languages, letters change form depending in which position they are found, which make transposition errors less likely because they would necessarily involve changing the form of the letter too which is a key confound (see the work of Friedmann). Even in German, the first letter of nouns is capitalised which is a confound in that it is unclear whether effects can be attributed to the position of the letter per se or the fact that first letters appear in capital form, and thus stand out. I think that given the above, the authors should consider whether it’s worth emphasising somewhere in their manuscript that these results provide support to theories mostly developed with English in mind, without necessarily extending to all other languages.

Author Response

We would like to thank the reviewer for their helpful comments – we have tried to work through each of the suggestions as best we can. - below you will see our responses in italics.

 1) Pages 5 and 6 (lines 201-209)

The description of the (non)word stimuli led to confusion. It took me a few re-reads till I understood that for each of the type (a) transposition nonword foils, there was a control nonword foil (type (b)) which was a letter string containing letters that were visually similar to the transposed letters in the type (a) items. At least, this is what I understood, but I am still not sure, Actually, the tables with results on subsequent pages helped clarify the manipulations. I would recommend adding a table to present the experimental items with examples on page 5/6.

This is an excellent point, and so we have added an additional table (see Table 1) in this section to hopefully aid clarity.

2) Page 6 (lines 213)

It is good practice to specify what software (and version) is used in experiments.

Apologies – the software used was Psychopy version 3.0.1 – and this has been added to the text – highlighted in yellow page 5.

3) Page 6 (lines 232-233)

The authors write: “To mitigate ceiling and floor effects, we applied a log-linear correction to both hit and false alarm rates [32]”. I do not understand how merely transforming the data can “mitigate ceiling and floor effects”, beyond just compressing the data to reduce skew which is the most common use of the log transformation as far as I am aware. Could the authors explain further?

This was merely to deal with the fact that if the observed score = 100% (that is no errors on hits or false alarms, which was the case for a the condition 1+2 for KL on our first testing observation) the z-transform becomes infinite and so you can’t compute a value for d – but we agree the use of “ceiling” here is clumsy so have removed the sentence in question.

4) Page 6 (lines 243-250)

The paragraph where the modelling approach is explained led to confusion. Firstly, the paragraph starts by mentioning “To investigate the effect of session…”. For one thing, differences between the five testing sessions (line 211-212) are not linked to the hypotheses the authors formulated, so it comes a bit out of the blue for the reader. For another thing, the authors do not actually include session as a predictor in their formula later on (line 250). Instead, they have included condition, which I guess stands for the letter transposition manipulation (i.e. 1+2, 2+3, 3+4, 4+5). This is what I could infer based on the Table results. The Table results also suggest that another important factor to include in models is Foil Type. Yet, this is also not included in models, much like Session. Finally, a related issue emerges later on page 8 (lines 290-291) where the authors write “...the effect of session (test condition)…”. This causes even more confusion as to what the terms “session” and “condition” actually stand for, and why the authors chose to use these terms interchangeably (see also lines 320-321), if they refer to different experimental aspects, as one would expect. The authors should clarify the above to resolve confusion.

Again sorry about the confusion – firstly it should read ‘condition’ (which is foil type) not ‘session’, this confusion arises because the four different conditions for the lexical decision experiments (i.e., 1+2, 2+3 and so on) were presented across different test session time periods (closely together) see lines 212 and 213 of the methods. In any case, we have removed all mention of the word session to keep the clarity of the condition. Moreover, we made things even more confusing by using the word ‘session’ for the fact KL was tested across two different occasions (18 months apart), so we have used the word ‘occasion’ to clarify this difference. Again, apologies for this, the changes are marked in yellow. We hope these changes along with the additional table make things clearer.

5) Page 6 (line 250)

The choice of a generalised liner model for the type of data presently reported is inappropriate. Specifically, the glm model violates the statistical assumption of independence of observations as the authors have repeated measurements from the same participants. A glmer, for example, would avoid this problem if the authors decided to analyse both the KL and GS data together, and have “participant” as a random effect. Yet, in practice, such a model would again fail most likely, as between participant variance cannot be effectively estimated with such a small sample size (N=2). In my experience, when dealing with case studies, statistical analyses are not practically feasible, and even if one manages to get the models to run, SE values would be large enough to cast doubt on the reliability/precision of model estimates. A more common approach when dealing with case studies is to report degree of deviation from normative values, either using assessments standardised based on the performance of healthy controls or by recruiting an age-matched control group as a reference point. Given that the study presently reported does not include the above, I can’t think of other statistical analysis suggestions I could propose. Thus, I would advise the authors to focus on descriptive stats (i.e. central tendencies such as mean accuracy) and comment on the absence of statistical analyses and normative values in a Limitations section.

Again we have been very guilty of a lack of clarity here and can only apologise – put simply we wanted an analysis approach, that would ask does accuracy differ across the four conditions for this particular patient/case in this particular observation? Control participants, in our experience are unlikely to make many, if any errors on this task (a forthcoming QJEP paper by us can illustrate this and see also original paper by Shalev and colleagues) so their inclusion here wouldn’t really help and asks a somewhat different question in our view (namely is condition X below a ‘cut-off’ of some such). Our aim here, which again wasn’t clear (we apologise), was to bring into focus the striking relative dissociation pattern across conditions for each particular patient/case. For example, on the first occasion of testing with KL, this pattern was so striking as to be self-evident - namely, he is perfect in his performance for the 1+2 condition relative to the other conditions (which was unclear in the previous presentation of the work and should be much clearer following your helpful suggestion of adding accuracy info to tables, see below). We explored that further with our non-parametric bootstrapping approach by considering estimated confidence intervals.  On occasion 2 with KL (and with the other patient) the 1+2 condition was not at ceiling, so we sought to try (in addition the bootstrapping approach) some other illustrative statistical approach and selected a straightforward logistic regression, to investigate the pattern a bit further – to be clear again we wanted the model to be estimating differences across the four conditions within participant. As a consequence, we were perhaps not clear that the estimate is independent in the condition sense that we interpreted it, because each condition is it is own particular observation in time. Though we do acknowledge that trials within conditions from a single participant are not strictly independent (e.g., item-level variance); but as you point out any alternatives across cases, etc without random effects is obviously not an option either. So, we do acknowledge that we likely deserve a slap on the wrist for this, around issues of assumptions of independence (ironically the original paper by Shalev and colleagues used Ch-squared, but this is no better for the same reasons you have made). Nonetheless, we are still somewhat keen to at least pragmatically retain this additional analysis. On advice, we sought to determine the degree to which the kind of level of impact the issues you raised could be identified – and as a consequence, we examined our original logistic regression models for overdispersion, which occurs when the variance in responses exceeds the level expected under a binomial model and can indicate dependence between observations. We computed the ratio of residual deviance to residual degrees of freedom for each analysis. The results were: KL (Experiment 1, Session 2) = 1.09; GS (Experiment 1, Session 1) = 1.27; KL (Experiment 2, Session 1) = 1.04; KL (Experiment 2, Session 2) = 0.96. A value of 1 indicates no overdispersion, and values greater than 1.5–2 are typically taken as evidence of serious model misfit (Collett, D. (2002). Modelling binary data. CRC press.). Since only GS Session 1 showed mild overdispersion, with all other models very close to 1. As a robustness check, again on advice, we re-fit the model for this set of observations using a quasibinomial specification, which estimates a dispersion parameter and adjusts the standard errors accordingly; and then compared estimates to that of the original log regression model. Importantly, the regression coefficients and statistical conclusions remained unchanged (all of this is in the logistic model R code on OSF). On that basis, we would conclude that our results are not likely meaningfully affected by violations of the independence assumption you have mentioned. However, we do acknowledge your challenges and the related issues in a new limitations section, and again stress the objective of the analyses were illustrative to complement observations from the bootstrapping approach. Nonetheless, you as reviewer may still feel this is inappropriate and if so, we will respectfully remove this analytical approach entirely.

6) Tables 1-4

The authors could include another column in the Tables to report on mean accuracy (e.g. in %). This information is currently embedded in the text or is not reported (e.g. line 287 “accuracy was not perfect”). It would be better to have all relevant performance measures presented together in each Table.

Again, this is an excellent idea we have added into each column accuracy for each condition – and acknowledge this has also caused some lack of clarity earlier. Again, apologies for this oversight.

7) Page 8 (lines 296-301)

The following comment may become irrelevant if the authors remove the statistical analyses as I advised them in my comment 5). If that is the case, please ignore the below.

In lines 296-301, the authors report statistical analyses results for the performance of KL in the second testing period. Yet, they do not do the same on the previous page (page 7) for the first testing period. The authors should report both stat results and if there is any inconsistency in outcomes, they should comment on this in the Discussion section.

See also response to comment 5 above – essentially in our first testing occasion with KL his accuracy was basically strikingly at ceiling (100%) for the key condition of interest (1+2) relative to others; this was perhaps less than clear because we used d-primes but have added % accuracy in the tables to assist. Since accuracy was so good and the key dissociation so self-evident in this first testing session, we simply focused on the obvious illustration of the clear lack of CI overlap for this condition (1+2) versus the others. In the second occasion of testing (18 months later) condition 1+2 was no longer perfect in accuracy – and although the similar bootstrap suggests a continuing advantage for the key condition over others (1+2) we had sought to see if we could undertake some subsequent analysis – to be clear (since it is evident we haven’t been), the idea here isn’t to demonstrate some difference relative to a control population per se, it is to demonstrate a within patient comparative advantage – that is an effect for them manifest as a function of trial type – we have explained above that we think the risk of dispersion issues are likely to be small, despite the potential impact of the independence assumption issue you quite rightly raised – so at this point we would again suggest one of two options, either we keep the analysis in as before and raise the technical points you have done in the limitations section (which we have done – see limitations section) – or we remove all of them entirely and just stick to reporting the bootstrapping simulation observations. We will leave this up to your discretion and will follow your best advice.

Additionally, in their supplementary materials uploaded on OSF (https://osf.io/f7che/) the authors have included their R code for the results reported on page 10 (lines 383-389), but not for the ones reported on Page 8 (lines 296-301) or for the other statistical results reported in the case of GS (lines 318-328). Please upload all scripts for replicability.

We have been back over what we uploaded and have made certain all the files are uploaded – you should be able to find the two main programs – that is, separate R code to create the bootstrap simulations and to undertake the regression, obviously they need a slight change depending on which file you are analysing. In each data folder (e.g., GS, KL etc) there are two different files one for the former program (bootstrap) and one for the regression. I hope that is clear. 

8) Page 9 (lines 324-328)

The two sentences in lines 324-328 essentially repeat the same information. I would remove one of them.

Thank you for this advice and this has been removed.

9) Page 12 (lines 483-484)

Finally, I found the introductory section and the discussion to be well-researched and well-written. The subsection “Implications and Future Directions” included the question “Would these effects generalise to languages with different orthographic directions?”. This is a key question, but I think there are more outstanding questions to address that should be mentioned. For one thing, the lack of a healthy control sample to which the performance of KL and GS can be compared is a limitation of this study, and should be addressed in a Limitations section, as mentioned in my comment 5). For another thing, beyond different orthographic directions in other languages, there is also other variability across writing systems that, in my view at least, raise questions regarding the “universality” of the first-letter advantage reflecting “a more abstract positional status” (line 416). While the attribution of the observed results to positional status may hold for English, it is unclear to me that the same could be said about typologically distant languages. For instance, in Semitic languages, letters change form depending in which position they are found, which make transposition errors less likely because they would necessarily involve changing the form of the letter too which is a key confound (see the work of Friedmann). Even in German, the first letter of nouns is capitalised which is a confound in that it is unclear whether effects can be attributed to the position of the letter per se or the fact that first letters appear in capital form, and thus stand out. I think that given the above, the authors should consider whether it’s worth emphasising somewhere in their manuscript that these results provide support to theories mostly developed with English in mind, without necessarily extending to all other languages.

Again, these are all excellent points – we have on advice included a limitations section in the General Discussion and have added in the very helpful points you have raised. I hope we have done them justice.

Reviewer 2 Report

Comments and Suggestions for Authors

Q1. The abstract contains several stylistic and formatting issues, such as misplaced punctuation ("..") and the inclusion of a reference citation, which should not appear in the abstract. Furthermore, the case study subjects (GK and FL) are introduced without any contextual explanation. Please clarify that these are pseudonyms for the two individuals under investigation, to avoid confusion with abbreviations or methods.

Q2. This is not an original research article in the traditional sense, but rather a report of two extended case studies. While the detailed qualitative description is appreciated, a sample size of two does not justify categorical or generalizable claims. The manuscript lacks sufficient information on how the diagnosis was established, who performed it, and under what conditions. I strongly recommend revising the title to clearly indicate that this is a case study report involving two individuals with a specific diagnosis, rather than presenting it as a generalizable clinical or cognitive study.

Q3. In the introduction, the sentence: “a neurodegenerative syndrome characterised by severe attentional deficits?” is written as a question. This seems a typo in the context of a scientific definition. Please rephrase it as a declarative sentence.

Q4. Recent work such as “Dual-System Recommendation Architecture for Adaptive Reading Intervention Platform for Dyslexic Learners” has introduced adaptive models to quantify lexical processing and learning using personalized word/pseudo-word recommendations. Although you are not expected to implement such a system, its absence represents a limitation in the current framework. We suggest briefly acknowledging this work as a potential avenue for future research or as a limitation in quantifying subject-specific linguistic dynamics.

Q5. In the Results section, please use clear subsection formatting to separate the findings for each individual subject. This will help structure the narrative and facilitate comparative analysis.

Note: This part is actually better written than the Methods section — good job on clarity and narrative structure here..

Q6. The explanation of word transpositions is confusing. For instance: (1 = 1+2 transposition, 2 = 2+3, etc.) is unclear to the reader without examples. Please include specific examples in the Methods section to illustrate each type of transposition.

Q7. (Not important) The manuscript would benefit from a more thorough and updated literature review. In particular, references to recent work published in Brain Sciences are absent. Including relevant citations from this journal could both strengthen your framing and show alignment with the scope of the journal...

Q8. In the Results section, you report means and perform comparative analysis across several linguistic variables. However, it is unclear whether the assumptions underlying these comparisons are met. For example, if any inferential tests are applied (even descriptively), the normality of the data should be evaluated — using tests such as Shapiro–Wilk or Anderson–Darling, especially given the small sample size! Please clarify whether these tests were used and justify the validity of any statistical procedures employed.

Q9. The Methods section lacks sufficient detail to allow replication... How were stimuli constructed, presented, and scored?

Q10. I think that Discussion section should be expanded to address the limitations inherent to n=2 designs and reflect more cautiously on generalization.

Q11. Ethical approval and informed consent should be explicitly stated (even for case studies).

I marked this submission as Major Revision, but I want to emphasize that I really liked your work. The case studies are very interesting and raise important questions in the field. I strongly encourage you to address the issues I outlined in the review — particularly regarding structure, clarity, and framing — as I believe your manuscript has strong potential and is definitely worth publishing with the appropriate revisions. Best regards to the team — I look forward to seeing your revised version!!!

Author Response

We would like to thank the reviewer for their helpful comments – we have tried to work through each of the suggestions as best we can. - below you will see our responses in italics.

Q1. The abstract contains several stylistic and formatting issues, such as misplaced punctuation ("..") and the inclusion of a reference citation, which should not appear in the abstract. Furthermore, the case study subjects (GK and FL) are introduced without any contextual explanation. Please clarify that these are pseudonyms for the two individuals under investigation, to avoid confusion with abbreviations or methods.

Thank you, we have removed the reference citation in the abstract – and removed the mention of GK and FL as these are the patients described in the earlier study cited – again, sorry this appears to have confused you because of our lack of clarity.

Q2. This is not an original research article in the traditional sense, but rather a report of two extended case studies. While the detailed qualitative description is appreciated, a sample size of two does not justify categorical or generalizable claims. The manuscript lacks sufficient information on how the diagnosis was established, who performed it, and under what conditions. I strongly recommend revising the title to clearly indicate that this is a case study report involving two individuals with a specific diagnosis, rather than presenting it as a generalizable clinical or cognitive study.

We are somewhat reluctant to do this as we rather like the title and respectfully disagree that is unclear. We do however agree that we should have added in some information about how PCA diagnosis is undertaken and how that relates to our patients, so have included this in the methods section (case descriptions), text highlighted in yellow.

Q3. In the introduction, the sentence: “a neurodegenerative syndrome characterised by severe attentional deficits?” is written as a question. This seems a typo in the context of a scientific definition. Please rephrase it as a declarative sentence.

Thank you for pointing this out we have made the change see yellow highlighted text.

Q4. Recent work such as “Dual-System Recommendation Architecture for Adaptive Reading Intervention Platform for Dyslexic Learners” has introduced adaptive models to quantify lexical processing and learning using personalized word/pseudo-word recommendations. Although you are not expected to implement such a system, its absence represents a limitation in the current framework. We suggest briefly acknowledging this work as a potential avenue for future research or as a limitation in quantifying subject-specific linguistic dynamics.

Thank you for this advice – we have since added a limitations section and included the point you have made – we hope we have done it justice.

Q5. In the Results section, please use clear subsection formatting to separate the findings for each individual subject. This will help structure the narrative and facilitate comparative analysis.

Note: This part is actually better written than the Methods section — good job on clarity and narrative structure here..

Again we apologise for the lack of clarity – we have now since added in some extra subheadings for the results section and these are all highlighted in yellow. Thank you for this advice.

Q6. The explanation of word transpositions is confusing. For instance: (1 = 1+2 transposition, 2 = 2+3, etc.) is unclear to the reader without examples. Please include specific examples in the Methods section to illustrate each type of transposition.

This is a good point – and similar to a comment made by Reviewer 1 – so we have now added a Table (see Table 1) in the methods section presenting a visualisation of examples relating to each experimental condition – thanks for this advice and we hope this is clearer.

Q7. (Not important) The manuscript would benefit from a more thorough and updated literature review. In particular, references to recent work published in Brain Sciences are absent. Including relevant citations from this journal could both strengthen your framing and show alignment with the scope of the journal...

Thanks – we had a look but were unable to find any obvious examples but remain open minded if you may have ideas.

Q8. In the Results section, you report means and perform comparative analysis across several linguistic variables. However, it is unclear whether the assumptions underlying these comparisons are met. For example, if any inferential tests are applied (even descriptively), the normality of the data should be evaluated — using tests such as Shapiro–Wilk or Anderson–Darling, especially given the small sample size! Please clarify whether these tests were used and justify the validity of any statistical procedures employed.

Again we apologise for the lack of clarity here - our inferential analyses were conducted at the trial level using binomial logistic regression, which does not assume normality of the response or residuals. For the signal-detection measures (d′ and c), we reported non-parametric bootstrap 95% CIs, which also do not rely on normality. Nonetheless, as reviewer 1 has pointed out, because the data are repeated trials within a single participant, we acknowledge that a key diagnostics concern likely relates to extra-binomial variation (overdispersion) rather than normality. As mentioned above, we therefore examined dispersion (residual deviance/df): KL-Exp1-Session2 = 1.09; GS-Exp1-Session1 = 1.27; KL-Exp2-Session1 = 1.04; KL-Exp2-Session2 = 0.96. Only GS-Session1 showed mild overdispersion, but re-fitting with a quasi-binomial model (which inflates SEs by the estimated dispersion) left coefficients and conclusions unchanged. In sum then, and we apologise for the confusion we have created here through lack of clarity, given these model families and checks, normality tests such as Shapiro–Wilk/Anderson–Darling are not applicable, and our procedures remain valid.

Q9. The Methods section lacks sufficient detail to allow replication... How were stimuli constructed, presented, and scored?

The stimuli were directly taken from the original study by Shalev and colleagues (first citation) in the work they undertook with their two attentional dyslexia patients. We are sorry this wasn’t clear and have added a sentence in the beginning of the methods section (highlighted in yellow) to reflect this. We have also amended the sentence (as per Reviewer 1) on the software used and the process of data collection (also highlighted in yellow in the methods section) which also computed accuracy for each item presented. We hope this now makes things clearer.

Q10. I think that Discussion section should be expanded to address the limitations inherent to n=2 designs and reflect more cautiously on generalization.

A similar point was made by Reviewer 1 and 3 so we have added this point in our new limitations section – and thank you for this advice.

Q11. Ethical approval and informed consent should be explicitly stated (even for case studies).

Again, we apologise for the lack of clarity here. We did say in the later - Institutional Review Board Statement: The study was conducted in accordance with the Declaration of Helsinki and approved by an IRAS Ethics committee (Project ID: 234933). Informed Consent Statement: Informed consent was obtained from all participants involved in the study. – but we have added a sentence to the case descriptions section in addition to this (highlighted in yellow).

I marked this submission as Major Revision, but I want to emphasize that I really liked your work. The case studies are very interesting and raise important questions in the field. I strongly encourage you to address the issues I outlined in the review — particularly regarding structure, clarity, and framing — as I believe your manuscript has strong potential and is definitely worth publishing with the appropriate revisions. Best regards to the team — I look forward to seeing your revised version!!!

Thanks to you for your helpful advice and kind words!

Reviewer 3 Report

Comments and Suggestions for Authors

There is insufficient clarity in the presentation of the study design.  Even though the manuscript states that two PCA patients participated in lexical decision tasks, the description is disorganised and necessitates close reading to fully comprehend the precise order of the tests, conditions, and experiments.  The justification for simplifying conditions in Experiment 2 (only 1+2 vs. 4+5 transpositions) could be clarified, and the distinctions between Experiment 1 (canonical format) and Experiment 2 (vertical format) are only partially explained.  Readers would find it easier to follow the methodology if there was a schematic overview (e.g., timeline of sessions, conditions tested, number of trials).

Although the study is based on only two PCA patients, the number of participants is not specified. Readers may assume a different study design or a larger sample as a result of this omission. Moreover, no numerical values or effect sizes are provided, despite the abstract's references to "significantly higher accuracy" and "robust effects." The precise number of participants as well as at least one or two significant numerical results should be included for transparency and clarity.
Update the abstract to include important quantitative results and to explicitly indicate the number of participants.

Furthermore, there is no systematic presentation of the correlations or comparisons between various experiments and time points.  The manuscript would benefit from a clearer structure that directly compares performance across experiments and explicitly highlights convergences and divergences, even though individual results for each patient are described in detail.  What can we learn about orientation invariance, for example, from the within-subject comparison between vertical and canonical formats?  Are the outcomes the same for both patients, and if not, what does this mean?

The limitations of the study are also not adequately addressed in the discussion section.  The findings' generalisability is severely limited by the extremely small sample size (two case studies), which needs to be more clearly acknowledged.

Include brief graphical summaries of the findings along with illustrative figures of the stimuli. The study's communicability and transparency would both be enhanced by these additions.

Author Response

We would like to thank the reviewer for their helpful comments – we have tried to work through each of the suggestions as best we can.

There is insufficient clarity in the presentation of the study design.  Even though the manuscript states that two PCA patients participated in lexical decision tasks, the description is disorganised and necessitates close reading to fully comprehend the precise order of the tests, conditions, and experiments.  The justification for simplifying conditions in Experiment 2 (only 1+2 vs. 4+5 transpositions) could be clarified, and the distinctions between Experiment 1 (canonical format) and Experiment 2 (vertical format) are only partially explained.  

We thank the reviewer for their helpful advice here – and it mirrors some of the earlier comments made by other reviewers. We have now included a table to walk through the conditions, and clarified the confusion raised by using the word ‘session’ in particular ways. In addition in relation to Experiment 2 and our use of a vertical format, we agree more information should be provided so have done so on pages 4 and 9 (see highlighted text) – we have just had accepted for publication a paper on PCA and non-canonical format presentation in another MDPI journal (Reports) which partly gave us the inspiration for exploring this manipulation, since changing the presentation format can have an impact on attentional processes in reading. So we have cited that paper now.

Although the study is based on only two PCA patients, the number of participants is not specified. Readers may assume a different study design or a larger sample as a result of this omission. Moreover, no numerical values or effect sizes are provided, despite the abstract's references to "significantly higher accuracy" and "robust effects." The precise number of participants as well as at least one or two significant numerical results should be included for transparency and clarity.
Update the abstract to include important quantitative results and to explicitly indicate the number of participants.

We are a little confused about some of these statements – in the original abstract it is written that we are replicating and extending this work in “two patients” with PCA, but have made some changes to the abstract which we hope are clearer. All analyses include z-scores so effect sizes are apparent, and the inclusion of accuracy scores should also make the dissociations in question (i.e., condition 1+2 vs others) much more transparent. We hope this assists in the requests you have made.

Furthermore, there is no systematic presentation of the correlations or comparisons between various experiments and time points.  The manuscript would benefit from a clearer structure that directly compares performance across experiments and explicitly highlights convergences and divergences, even though individual results for each patient are described in detail.  What can we learn about orientation invariance, for example, from the within-subject comparison between vertical and canonical formats?  Are the outcomes the same for both patients, and if not, what does this mean?

Again I think we haven’t been clear in the motivations of the work – so have tried to be so in the changes we made in response to the other two reviewers also. The idea isn’t to directly contrast different patients or contrast the test – retest occasions undertaken, the idea is to demonstrate that all patients at the times of observations are showing a striking performance (within participant) of relatively higher performance on a particular condition of interest. For example, in our first testing occasion with KL, his accuracy was basically strikingly at ceiling (100%) for the key condition of interest (1+2) relative to others – and our subsequent observations found this relative advantage remains over time and even in non-canonical presentation situations – again to be clear (since it is evident we haven’t been), the idea here isn’t to demonstrate some difference relative to a control population per se, it is to demonstrate a within patient comparative advantage – that is an effect for them manifest as a function of trial type.

The limitations of the study are also not adequately addressed in the discussion section.  The findings' generalisability is severely limited by the extremely small sample size (two case studies), which needs to be more clearly acknowledged.

Many of these comments are similar to those made by the previous reviewers and we certainly acknowledge these challenges – we have since created a “Limitations” section in the General Discussion that covers the issues you have raised and others. We hope that what we have added reflects the spirit of your helpful feedback.

Include brief graphical summaries of the findings along with illustrative figures of the stimuli. The study's communicability and transparency would both be enhanced by these additions.

On the advice of other reviewers, we have now included the accuracy information in all results tables - we feel making a bar graph of exactly the same information would be somewhat redundant and create over duplication - we hope this meets your needs. At the same time a table (see Table 1) has now been added as you have suggested to illustrate a key example of each of the critical conditions. Thanks for your advice.

Round 2

Reviewer 2 Report

Comments and Suggestions for Authors

Q1. Solved.

Q2. This is not what I was expecting in either case. I still believe the title should explicitly indicate that the manuscript is based on two individual case studies.

Additionally, regarding the description of the patients' diagnoses, I was expecting a more specific explanation — namely, what kind of specialists were involved in diagnosing these cases, where the assessments took place, and under what clinical conditions... This is essential and goes beyond simply referencing PCA as a condition.

Q3. Solved.

Q4. Please include the corresponding reference(s).

Q5. Solved.

Q6. Solved.

Q7. Here are some examples that might be relevant to your context (please ensure they are applicable before including them — do not cite them without reviewing their content first):

  • Spoonerism Beyond Language: A Multi-Componential Perspective on Phonological Awareness

  • Discrimination and Integration of Phonological Features in Children with Autism Spectrum Disorder: An Exploratory Multi-Feature Oddball Protocol

  • Exploring Links Between Lexical Representations and Cognitive Skills in School-Aged Children with High-Functioning Autism Spectrum Disorder

  • Lenition in L2 Spanish: The Impact of Study Abroad on Phonological Acquisition

Q8. I didn’t take this into account — you are right. Solved.

Q9. Solved.

Q10. Solved.

Q11. Solved.

Great work overall — you've substantially improved the manuscript. Keep it up!

Author Response

Q2. This is not what I was expecting in either case. I still believe the title should explicitly indicate that the manuscript is based on two individual case studies.

Additionally, regarding the description of the patients' diagnoses, I was expecting a more specific explanation — namely, what kind of specialists were involved in diagnosing these cases, where the assessments took place, and under what clinical conditions... This is essential and goes beyond simply referencing PCA as a condition.

Thanks for that we have now ammended the title under your suggestion - and we have also ammended the following sentence with reference to the diagnosis specilaists involved, hope that is ok - "In both our cases initial diagnosis was provided by a neurologist on the basis of the pattern of features described above, which included neurological, radiological and behavioural assessment by a older adult psychiatrist and a specialist neuropsychologist, at relevant hospital clinics." 

Q7. Here are some examples that might be relevant to your context (please ensure they are applicable before including them — do not cite them without reviewing their content first):

  • Spoonerism Beyond Language: A Multi-Componential Perspective on Phonological Awareness

  • Discrimination and Integration of Phonological Features in Children with Autism Spectrum Disorder: An Exploratory Multi-Feature Oddball Protocol

  • Exploring Links Between Lexical Representations and Cognitive Skills in School-Aged Children with High-Functioning Autism Spectrum Disorder

  • Lenition in L2 Spanish: The Impact of Study Abroad on Phonological Acquisition

thank you - most of these were weren't sure were relevant since they largely involved children - but we have used the last one as it is relevant to our discussion of generalisation to other languages - thanks again.
